# 3D printing of self-healing personalized liver models for surgical training and pre-operative planning

Yahui Lu[1,5], Xing Chen[2,3,5], Fang Han[2,3], Qian Zhao [1], Tao Xie [1], Jingjun Wu [1,4] ✉ & Yuhua Zhang [2,3] ✉

3D printing can produce intuitive, precise, and personalized anatomical models, providing invaluable support for precision medicine, particularly in areas like surgical training and preoperative planning. However, conventional 3D printed models are often significantly more rigid than human organs and cannot undergo repetitive resection, which severely restricts their clinical value. Here we report the stereolithographic 3D printing of personalized liver models based on physically crosslinked self-healing elastomers with liver-like softness. Benefiting from the short printing time, the highly individualized models can be fabricated immediately following enhanced CT examination. Leveraging the high-efficiency self-healing performance, these models support repetitive resection for optimal trace through a trial-and-error approach. At the preliminary explorative clinical trial (NCT06006338), a total of 5 participants are included for preoperative planning. The primary outcomes indicate that the negative surgery margins are achieved and the unforeseen injuries of vital vascular structures are avoided. The 3D printing of liver models can enhance the safety of hepatic surgery, demonstrating promising application value in clinical practice.

Liver surgery has entered a new era of precision after the intuitive and empirical phase[1]. Conventional computed tomography (CT) or magnetic resonance imaging (MRI) can no longer meet the demands of preoperative planning. Consequently, three-dimensional (3D) visualization has gained acceptance as an important tool for simulating complex spatial relationships between veins and biliary structures in a liver, as well as rendering tumor or liver volumes[2]. Despite these advantages, 3D visualization can only be demonstrated through a two-demission (2D) display and fails to reflect liver elastic modulus or provide tactile feedback for surgeons. Additionally, it is not capable of simulating liver deformation caused by mobilization and retraction of the liver, as well as effects of respiratory movements during surgery. Accordingly, 3D printed physical models of patient anatomies are being developed to overcome these obstacles, enabling personalized surgical training as well as preoperative planning[3].

As physical objects, the 3D printed models can be positioned in desired fashions to explore optimal surgical approaches during the preoperative planning process[4]. However, it is worth noting that existing 3D models utilize commercially available printing materials and do not take into consideration the texture and modulus of the liver[5,6]. Consequently, the models are unable to accurately reflect liver deformation caused by surgical manipulation and respiratory movements. Moreover, the 3D printed models should be utilized for repetitive practice before the actual operation, allowing surgeons to experiment with different surgical cutting planes. In order to meet the demand for high accuracy and the freedom to repetitively cut with a

[1]State Key Laboratory of Chemical Engineering, College of Chemical and Biological Engineering, Zhejiang University, Hangzhou 310027, China. [2]Zhejiang Cancer Hospital, Hangzhou, Zhejiang 310022, China. [3]Hangzhou Institute of Medicine (HIM), Chinese Academy of Sciences, Hangzhou, Zhejiang 310018, China. [4]Ningbo Innovation Center, Zhejiang University, Ningbo 315807, China. [5]These authors contributed equally: Yahui Lu, Xing Chen. ✉e-mail: jingjunwu@zju.edu.cn; zhangyuhua1013@126.com

scalpel blade while preserving structural integrity of the model, a combination of stereolithographic 3D printing and self-healing materials with liver-like modulus is expected to offer a feasible strategy with low cost and high efficiency.

In general, the self-healing capability can be achieved by introducing dynamic covalent or noncovalent interactions into the polymer network[7]. Among the various dynamic bonds, disulfide bond[8], Diels-Alder coupling[9,10], boronic ester bond[11], thiocarbamate bond[12], and ionic bond[13] are widely utilized and have recently been formulated into 3D printable resins for the fabrication of self-healing 3D objects. Although a decent healing efficiency (95%) can be obtained for a fractured sample, the high temperature (80 °C) and long healing time (12 h) are practically inapplicable for implement in medical condition. The mildest healing temperature reported for a stereolithographic printable polymer is 65 °C by incorporating the boronic ester bond, which is still too high. Ideally, self-healing should occur within several minutes at ambient temperature. Besides, it is preferred that the self-healing polymers is easy-printed and cost-effective.

One of the critical contradictories arises from the fact that the sufficient stable chemical crosslinking is favorable for reliable 3D printing but significantly decreases the self-healing efficiency due to the inhibition of the chain segment motion[9]. In previous studies, we have reported the 3D printing of physically crosslinked rigid plastics without introduction of any covalent crosslinker[14]. Comparing with the existing self-healing polymers with various dynamic covalent crosslinking, purely physically crosslinked polymers are more promising candidates for self-healing because the linear polymer chain segments are more prone to migration, diffusion, and entangling, which are favorable for the crack healing. Furthermore, the absence of covalent crosslinker also facilitates the modulation towards the ultra-low

modulus to mimic the mechanical properties of the liver. Herein, we report the 3D printing of personalized liver models based on the self-healing elastomers with liver-like softness, which are endowed by hydrogen bond interactions and linear polymer chain topology between 4-acryloylmorpholine (ACMO) and methoxy poly (ethylene glycol) acrylate (mPEGA). These models can be repetitively cut and offer a trial-and-error method for planning optimal surgical traces before surgery. The 3D printed liver models can be employed for surgical training and preoperative planning, improving the degree of certainty and enhancing the safety of hepatic surgery.

## Results

### Materials design and 3D printing

A top-down DLP process is chosen since it provides better printing stability (e.g., a good self-support for soft materials) in comparison to the bottom-up process, especially for large-sized objects. The patterned UV light is illuminated from the top as the printing platform gradually moves downward during the printing (Fig. 1a). The photo sensitive liquid resin consists of a rigid monomer ACMO, a soft monomer mPEGA, a photo-initiator Irgacure 819, and a light absorber Sudan III. The resin and corresponding elastomers are denoted as ACEG-x, where x represents the weight percentage of ACMO. The curing kinetics, which is calculated from the double bond conversion, is studied using an off-line real time-infrared analysis. As shown in Fig. 1b, the copolymerization proceeds at a fast rate and reaches gelation in approximately 10 s, deriving from the entanglement of the polymer chains. Molecular weight as large as 500 kg/mol is measured from the gel permeation chromatography, ensuring good mechanical properties of the resulted elastomers. Pillars with diameters of 500 μm can be readily printed, which is a quite high printing resolution

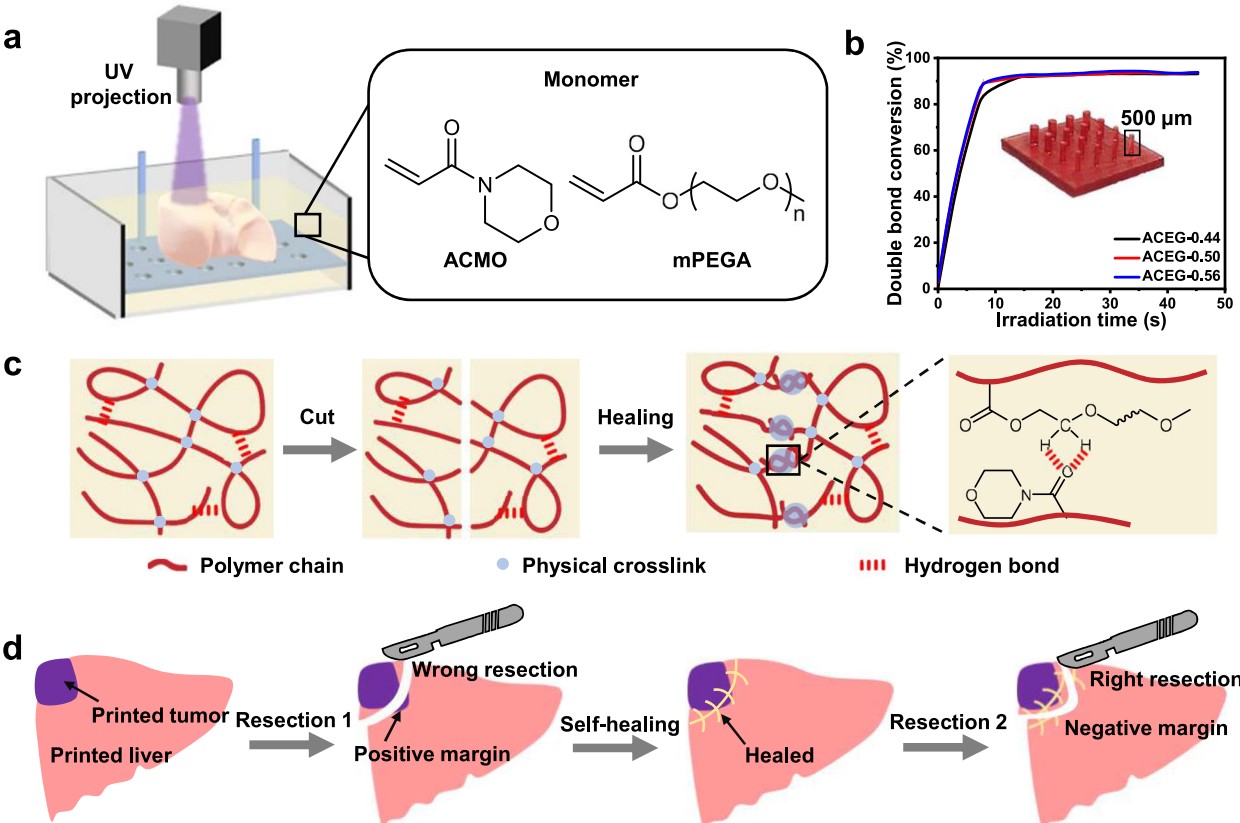

**Fig. 1 | Schematic of 3D printing self-healing elastomeric liver models. a** The setup of DLP 3D printing and the chemical structures of photocurable resin. **b** The curing kinetics of the ACEG resin with different ACMO/mPEGA ratios. The inserted image shows the printed pillars with different radius. **c** Schematic of the self-healing mechanism. **d** Schematic of self-healing elastomeric liver models for surgical training.

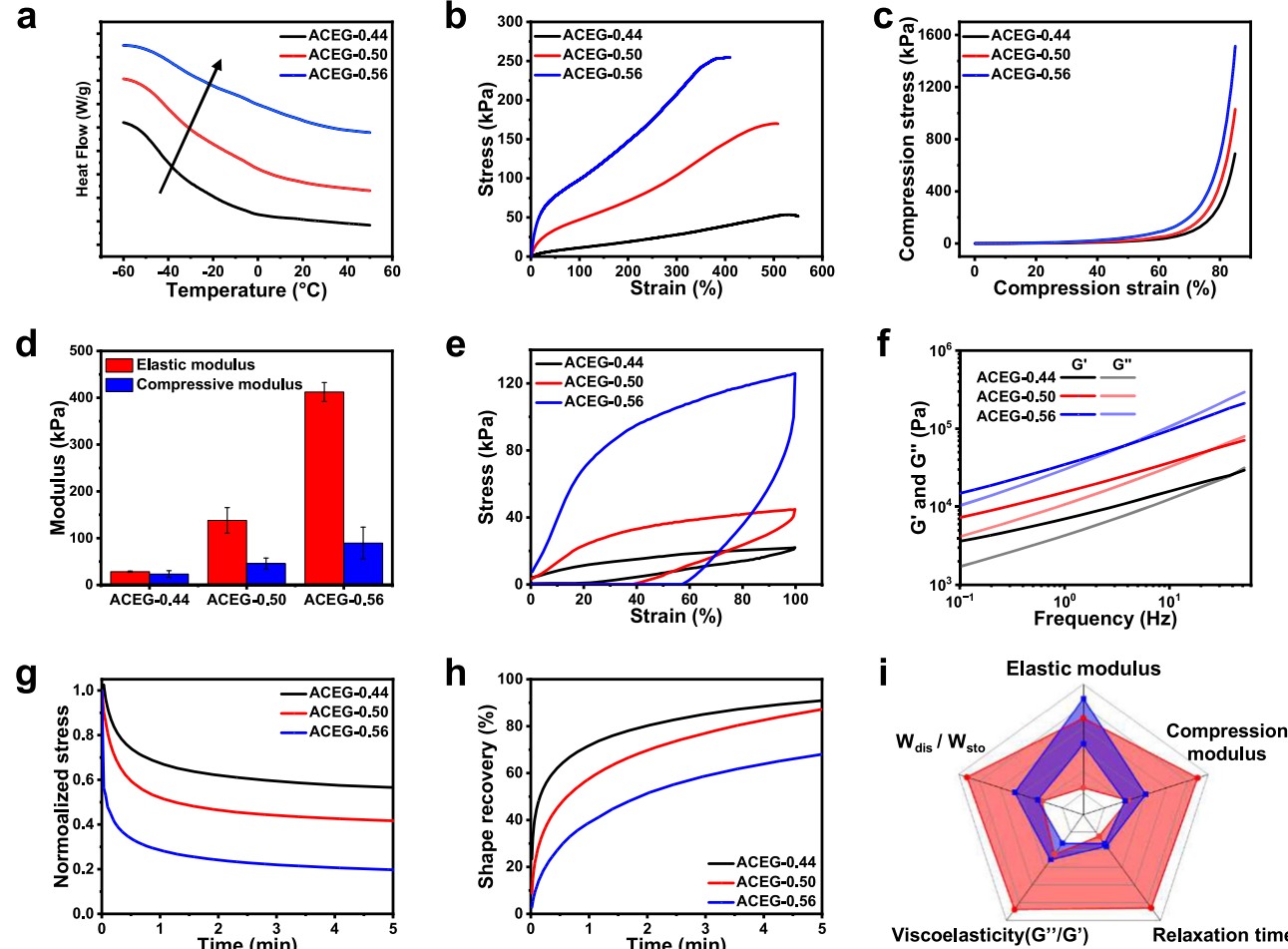

**Fig. 2 | Mechanical properties of the self-healing ACEG-x elastomers. a** DSC curves. **b** The tensile stress-strain curves (Strain rate: 100%/min). **c** The compressive stress-strain curves (Strain rate: 10%/min). **d** The tensile modulus and compressive modulus. Error bars represent standard deviation $n = 3$. **e** Hysteretic curves for dissipated to stored energy density (Strain rate: 100%/min). **f** Storage modulus and loss modulus measured by rotational rheology. **g** Iso-strain stress relaxation curves (strain: 20%). **h** Shape recovery curves (strain: 20%). **i** Comparison of mechanical properties between the self-healing elastomers (red) and human liver (blue). $W_{dis}$ (dissipated energy density) is the area between the loading and unloading curve, $W_{st}$ (stored energy density) is the area under the unloading curve.

considering the softness of the ACEG-x resin. The introduction of mPEGA not only decreases the modulus of the printed materials, but also forms abundant of intermolecular hydrogen bonding between methylene and amide groups. As a result, the ACEG-x resin shows both a good mechanical robustness and self-healing capability. In addition, the linear topology of ACEG-x elastomers benefits an easier diffusion and entanglement of the polymer chains, which also contributes to a fast room-temperature self-healing (Fig. 1c). By utilizing the above-mentioned printing process and elastomeric resin, we are able to 3D print a self-healable high-fidelity liver model which can be applied in surgical training and preoperative planning by repeatedly cut and self-healing (Fig. 1d).

**Mechanical properties and self-healing performance**

Mechanical properties of the ACEG-x elastomers vary accordingly with the composition. All the samples show a good softness, with $T_g$ values far below the room temperature (Fig. 2a). In the meantime, they possess an excellent stretchability with the strain-at-break above 400%. The tensile modulus and compressive modulus can be tuned from 28 kPa to 412 kPa and 23 kPa to 90 kPa, respectively (Fig. 2b–d). From the aspect of stiffness, the ACEG-x elastomers match well with the human liver[15]. Because the liver is a non-linear viscous material, the viscoelasticity of the ACEG-x elastomers is also evaluated extensively. It

can be observed from the hysteretic curves (Fig. 2e) that when the material is stretched, a noticeable energy dissipation can be observed, resulting from the sliding and friction of molecular chains. The dissipated energy increases as the ACMO content increases. Also, the energy dissipation is affected by stretching rate and strain (Supplementary Fig. 1). Rotating shear test shows that all samples exhibit elastic behavior ($G'' < G'$) at low frequency and viscous behavior ($G'' > G'$) at high frequency, which is a typical viscoelastic behavior (Fig. 2f). When the material is subjected to iso-strain stress relaxation and shape recovery experiment, the stress decay is faster and the shape recovery is slower with higher ACMO content (Fig. 2g, h). The tests simulate the deformation and recovery of the models during surgical training, and show that the models can recover relatively quickly after deformation. The viscoelastic properties of the ACEG elastomers obtained from the aforementioned mechanical and rheological tests are summarized in Fig. 2i, and compared with those of the human liver[16,17]. It can be observed that the ACEG elastomers can effectively mimic the biomechanical characteristics of the liver, which is crucial for its utilization in printing highly realistic liver models for surgery training and preoperative training.

ACEG-0.50 is chosen for further study because it exhibits a modulus of 140 kPa, which is similar to the modulus of liver. The self-healing properties of the ACEG-0.50 elastomer has been investigated

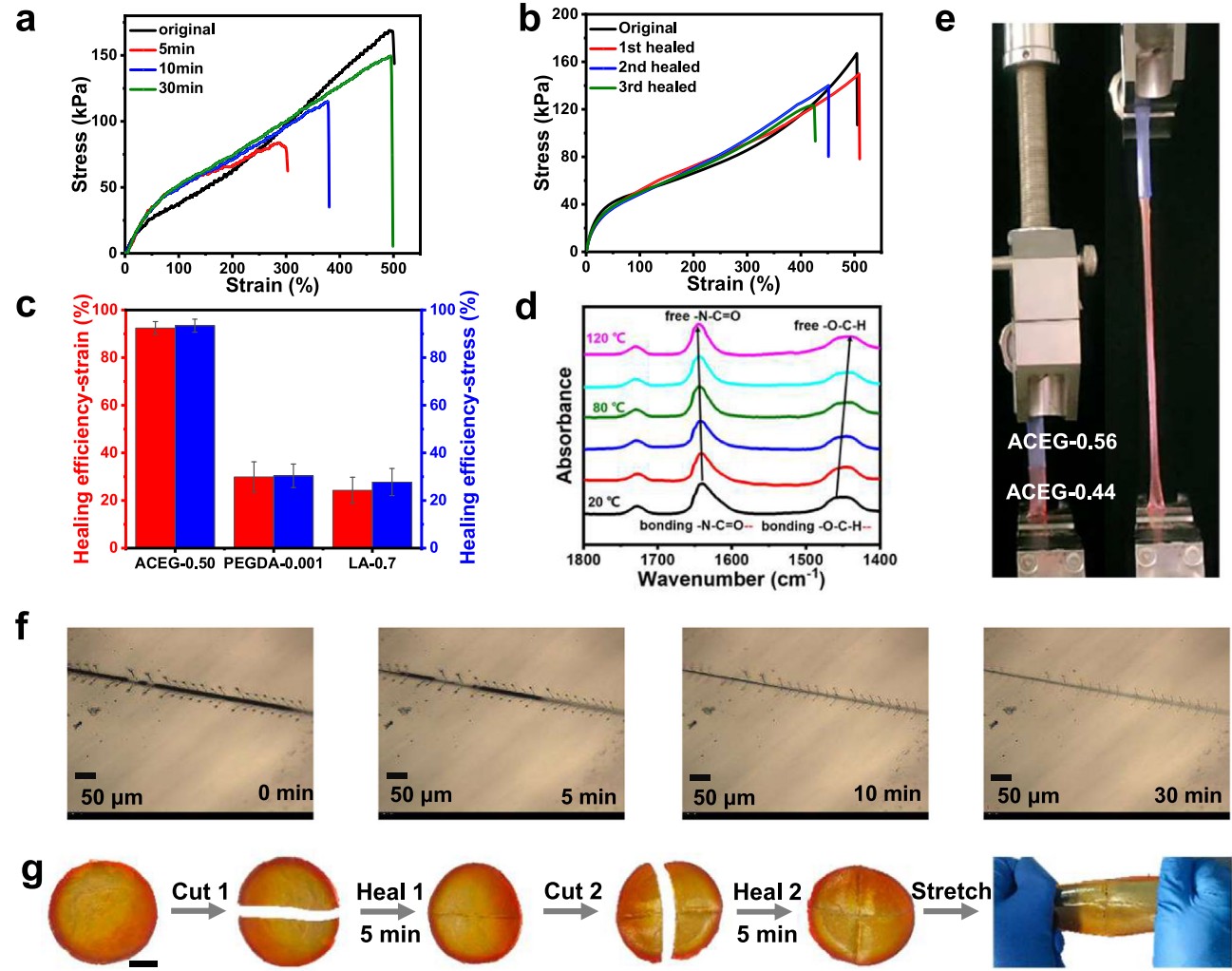

**Fig. 3 | Self-healing performance. a** Stress–strain curves of the self-healed samples with different healing time. **b** Multiple cutting-healing cycles of ACEG-0.50 sample. **c** Self-healing efficiency of different materials after being healed for 30 min. Error bars represent standard deviation $n = 3$. **d** Variable temperature FTIR spectra to illustrate the existence of hydrogen bond. **e** Self-welding of different ACEG elastomers. Strain rate: 100%/min. **f** The optical image of a scratch with different healing time. **g** Demonstration of a healed sample being deformed. Scale bar: 2 cm.

both qualitatively and quantitatively. The original samples are cut into two parts completely, and then the cutting surfaces are connected and healed under ambient conditions. The healing efficiency with respect of modulus, tensile strength, and strain at break can exceed 50% when the elastomer is healed for 5 min, and reaches 80% after 10 min. After being healed for 30 min, a similar stress-strain curve comparing to the original sample can be observed (Figs. 3a, c). In addition, there is no obvious decline in the healing efficiency after multiple cutting-healing cycles (Fig. 3b). The rapid self-healing firstly attributes to intermolecular hydrogen bonding, which is confirmed by variable temperature FTIR spectra. As shown in Fig. 3d, with increasing temperature from 20 to 120 °C, the absorption peak of the amide carbonyl group shifts from 1640 cm$^{-1}$ to 1644 cm$^{-1}$, while the methoxyl group at 1447 cm$^{-1}$ shifts to 1440 cm$^{-1}$. The shift indicates the existence of the hydrogen bonding between the methoxyl group and amide carbonyl group, which is formed at low temperature and broken at high temperature. When mPEGA is replaced by another soft acrylate monomer, lauryl acrylate (LA), the healing efficiency declines significantly under the same healing condition because of the absence of hydrogen bonding (Fig. 3c and Supplementary Fig. 2). Generally, the strength and healing efficiency for the existing self-healing materials with dynamic covalent bonds will gradually decrease with the increase of healing cycles, which is reasonably explained by the difficulty in fully

connect of the broken interface and the possible fracture of the permanent crosslinked network. In contrast, the ACEG elastomers without any chemical crosslinking undergo a sufficient polymer chain diffusion/entanglement, benefiting to the stable self-healing capacity. When tiny amounts of PEGDA (crosslinker, 0.1 wt%) are introduced, a dramatic decline of the self-healing efficiency can be seen (Fig. 3c and Supplementary Fig. 3). The excellent self-healing can also be conducted between ACEG elastomers with different compositions and mechanical properties as shown in Fig. 3e. So when preparing a liver model, we can individually print different parts, such as the tumor and the liver tissue with different materials, and then assemble them through self-welding. The multi-material liver models prepared by this strategy are closer to the true liver comparing to the models printed by a single material. For qualitative study, the healing process of a scratch on the ACEG-0.50 elastomer is recorded by an optical microscope. Figure 3f shows that the scratch (20 μm in width) heals automatically and leaves behind only a slight trace in about 30 min at 25 °C. Because of the non-chemically crosslinked characteristic of the ACEG elastomers, the polymer chains may flow at microscale, promoting the self-healing process. Owing to the excellent mechanical properties of the ACEG-0.50 elastomer, two pieces of samples can already undergo large deformation after being self-healed for only 5 min (Fig. 3g). Notably, although slight traces cannot be completely recovered, it will not bring

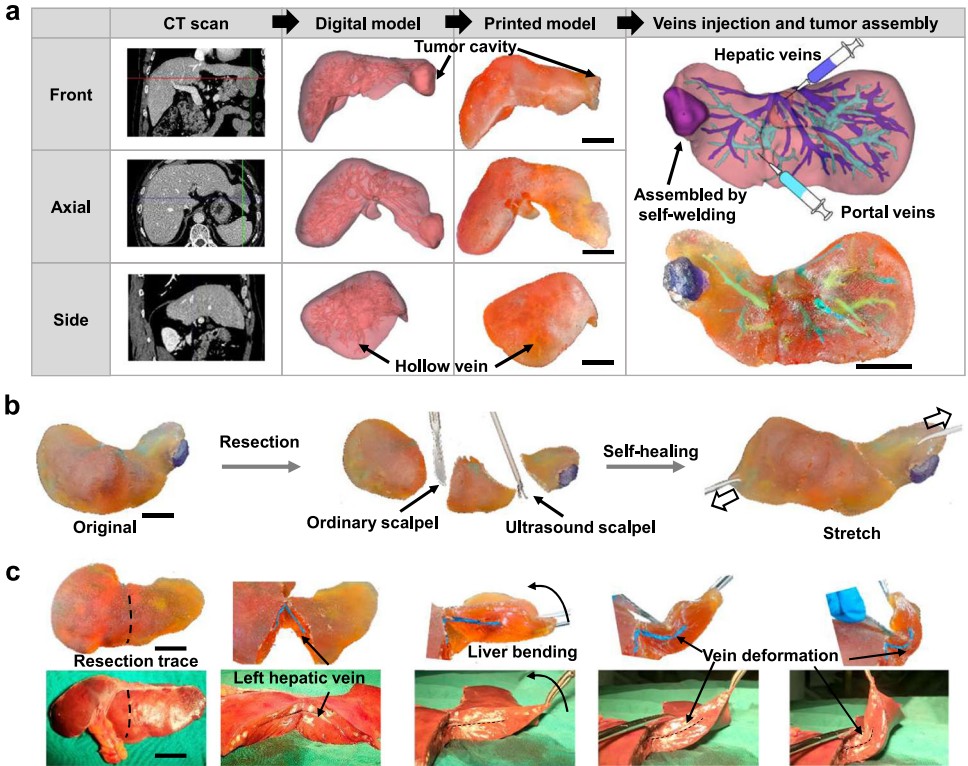

**Fig. 4 | Fabricating process of the liver models and the behavior of self-healing and deformation. a** Preoperative CT information, 3D digital, printed models, veins injection and tumor assembly of a tumor in segment VIII of the liver. **b** Self-healing behavior under the resection of ordinary scalpel and ultrasound scalpel. **c** Comparison of veins deformation between the printed model and human liver. Scale bar: 2 cm.

negative impact to the targeted applications of surgical training and preoperative planning.

### Fabricating process of the liver models

These data give us confidence that the 3D printed liver models using self-healing ACEG elastomers can be applied in clinical practice. After receiving an approval from the Institutional Ethical Review Board (IRB-2023-296), several personalized liver models are printed based on real data from typical patients. A typical fabrication process includes computed tomography scanning, digital model construction and 3D printing (Fig. 4a). Multi-layered 2D CT grayscale images are integrated into visualize 3D liver digital models. A liver model with a cavity reserved for a tumor and hollow veins is constructed on digital model by Boolean operation. A tumor printed by different materials is assembled on the cavity by self-welding. Hollow hepatic veins and portal veins are filled by blue paste and green paste, respectively. As the most commonly used cutting devices in the operating room, ordinary scalpel and ultrasound scalpel are tested on printed model (Fig. 4b). The material can be easily cut and the self-healing performance are similar in both conditions. The healed model can withstand various deformation, such as stretching, twisting and bending. Furthermore, a left half liver specimen removed from a patient is compared with the corresponding left half part of the printed model to investigate the effect of deformation behavior (Fig. 4c). We perform left lateral lobectomy both on the specimen and model, and the left hepatic vein is exposed. When the same external force is applied to the specimen and model, they show similar deformations, and the exposed veins also deform in an analogous way. It demonstrates that the models can simulate the deformation of liver itself as well as the anatomic structure inside, which are inevitably caused by surgical movements during an operation. This effect is difficult to achieve by other 3D visualization methods.

### Surgical training and preoperative planning

3D personalized liver models are used for surgical training in two typical liver surgical operations commonly performed in clinical practice. The first case shown in Fig. 5a presents a tumor with an outgrowth trend in segment VIII of the liver, necessitating an enucleation surgery. A novice doctor is tasked with practicing on the model to perform a competent resection, given their limited experience. Initially, a cutting trace immediately adjacent to the tumor boundary is planned, resulting in a positive surgical margin. It's widely accepted that a positive margin is associated with early recurrence in malignant liver disease[18,19]. Subsequently, the resected tumor is repositioned, and self-healing occurred within 5 min. To achieve a negative margin, the novice doctor extends the cutting trace outward by approximately 1 cm (Trace 2). However, this results in an injury to the right hepatic vein, which could pose a life-threatening risk to patients and necessitate a repair surgery (Resection 2). Subsequently, cutting trace 3 is recalibrated based on the prior two unsuccessful attempts, considering the rapid self-healing, and is positioned between Trace 1 and 2. A successful local resection of the tumor with enough negative margin and without injuring the right hepatic vein is achieved (Resection 3). Besides local hepatectomy, anatomic liver resection, which is performed along pre-existing structures, is often recommended[20]. Accurate identification of the appropriate cutting plane is crucial for performing anatomic hepatectomy. Figure 5b illustrates a candidate lesion for left hemi-hepatectomy in segment IVa of the liver. Performing the surgery along Trace 1 results in an injury to the middle hepatic vein, which serves as a crucial anatomical landmark for distinguishing left and right hemi-liver. Impairment of the middle hepatic vein can cause blockage of blood flow from the remaining liver tissue, leading to postoperative liver failure[21]. Conversely, resection guided by Trace 2 results in redundant liver tissue on the left side of the middle hepatic vein, which could potentially

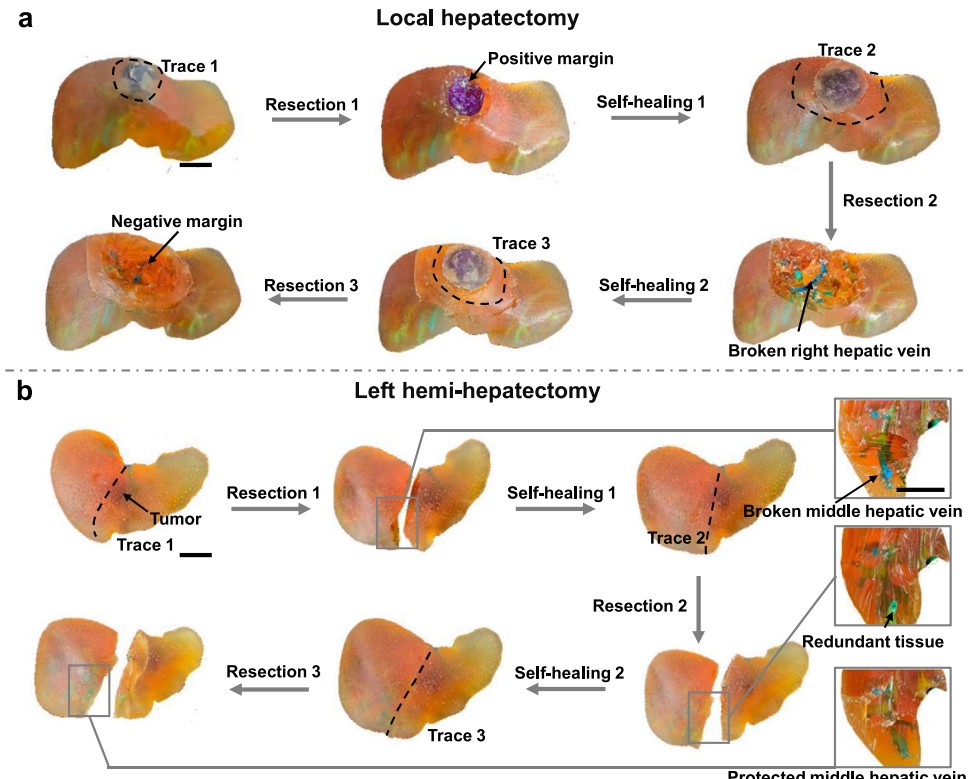

**Fig. 5 | Surgical training for two typical liver surgical operations. a** A local resection with negative margin and without an injury of right hepatic vein is performed after two times of resection and healing. **b** A standard left hemi-hepatectomy is carried out after two times of resection and healing. Scale bar: 2 cm.

overlook hidden metastases and increase the risk of recurrence. Subsequently, Trace 3 is established using the repeatedly self-healed model, ensuring the preservation of the middle hepatic vein according to this new trace. Another four local hepatectomy cases (Supplementary Figs. 4–7) and four left hemi-hepatectomy (Supplementary Figs. 8–11) cases with different shapes and pathological features are fabricated and implemented for surgery training, which helps novice doctors quickly accumulate clinical experience.

Benefiting from short fabricating time and low material cost, the highly personalized models can be printed soon after preoperative enhanced CT examination and the preoperative planning can be implemented on the model before the actual surgery. A phase 1, preliminary interventional study is registered on ClinicalTrials.gov (ID: NCT06006338). As illustrated in Fig. 6a (Patient 5), after establishing a reasonable resection by multiple iterations of tentative resections, a successful local hepatectomy on the patient is performed along the preoperative planning. Negative margin is achieved and the damage of right hepatic vein is avoided. Furthermore, for the planning of laparoscopic lateral segmentectomy as Fig. 6b (Patient 4), it is evitable to injury portal veins. So, it is determined to protect the trunk portal vein and break the end of two branches. Based on this expectation, a reasonable resection in real surgery is achieved, which is consistent with preoperative planning. A total of 5 cases with different pathological features are investigated as Supplementary Fig. 12 and the characteristics of patients as well as the clinical outcomes are detailed in Supplementary Table 1. No preoperative planning related adverse event was observed. Preoperative planning on the models can assists doctors to predict the possible outcomes of resection and avoid the unforeseen injuries of vital vascular structures. It maximizes the degree of certainty and minimizes the possibility of emergency situations during an operation in clinical practice, which is hard to be achieved by previously reported 3D printed models and other 3D visualization methods.

## Discussion

To fully meet the requirements of precision medicine in scenarios such as surgical training and preoperative planning, we report a facile and valid method for preparing personalized liver models by stereolithographic 3D printing with self-healing elastomers. The elastomers, synthesized through photo-initiated copolymerization of two inexpensive and commercially available monomers (ACMO and mPEGA), exhibit tunable modulus and efficient self-healing capability. Due to the linear polymer chain topology and hydrogen bond interactions, the 3D printed elastomers show a high self-healing efficiency within several minutes at ambient temperature. Benefiting from short fabricating time and low material cost, the highly personalized models can be printed soon after preoperative imaging examination. Taking advantage of their self-healing capabilities, a suitable cutting trace can be established after multiple iterations of tentative resections. By incorporating a model-based surgical approach into conventional preoperative planning, tumor-free surgical margins can be achieved while ensuring the protection of important anatomical structures. The 3D printed liver models with self-healing abilities might be a promising tool to improve operation safety and efficiency in the future clinical practice.

## Methods

### Ethics statement and clinical trials

The study protocol was reviewed and approved by the Institutional Ethical Review Board, Zhejiang Cancer Hospital (IRB-2023-296). Research process are conformed to the ethical standards for medical research involving human subjects. Participants provided written informed consent prior to taking part in the study. The consent explicitly conveyed that demographic and clinicopathological information would be utilized for academic research and eventual publication. No financial compensation was offered for participating in this study. The clinical trial (ClinicalTrials.gov ID: NCT06006338) is a phase

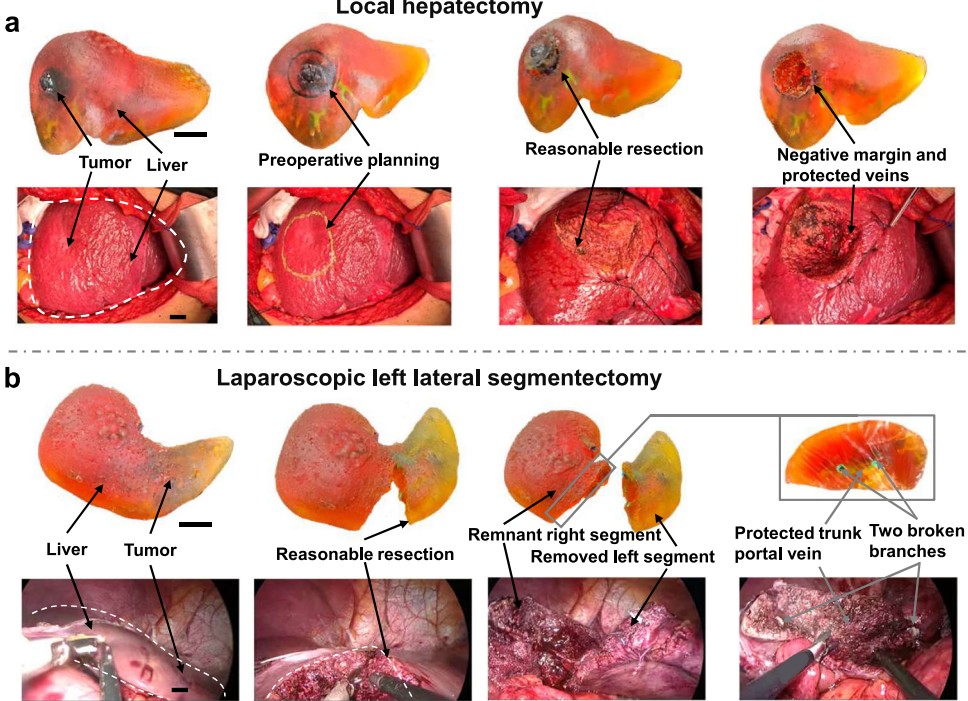

**Fig. 6 | Preoperative planning on models for two real surgeries in clinical. a** Local hepatectomy. **b** Laparoscopic left lateral segmentectomy. Scale bar: 2 cm.

1, preliminary interventional study activated on 23 Aug 2023. A total of 5 patients were enrolled for the study of preoperative planning, and the first case was investigated on 16 Sep 2023. The primary outcome indicates it can assist doctors achieve tumor-free surgical margins. The secondary outcomes show that doctors can avoid the unforeseen injuries of vital vascular structures after preoperative planning. It maximizes the degree of certainty and minimizes the possibility of emergency situations during an operation, enhancing the safety of hepatic operation in clinical practice.

## Materials

4-acryloylmorpholine (ACMO), methyl poly (ethylene glycol)600 acrylate (mPEGA 600) was purchased from Guangzhou Lihou Trading Corporation. Irgacure 819 was purchased from Macklin. Sudan III was purchased from Aladdin. Blue and green acrylic pastes were purchased from Marie's. All chemicals were used as obtained unless other noted.

## Synthesis of elastomers

The ACMO and mPEGA with different weight ratio were mixed together. 1 wt% of Irgacure 819 and 0.02 wt% of Sudan III were then dissolved in the mixture. The precursor was poured into a polytetrafluoroethylene mold with a thickness of 1 mm. The mold was conducted into a UV chamber (66 mW/cm², 265–700 nm) for 60 s.

## 3D printing

CT images were scanned by Siemens somatom Definition FlashDual Source CT (DSCT). Digital liver models were built by Yorktal Digital Medical Imaging and exported as STL format. A STL liver model with a cavity reserved for a tumor and hollow vein was constructed on digital model by Boolean operation. The STL liver models were 50% smaller than original size and printed by top-down 3D printer (Prismlab, RP-400). The STL tumor models were 50% smaller than original size and printed by a bottom-up 3D printer (Shining 3D, AccuFab-D1s) Liver and tumor were printed by ACEG-0.50 and ACEG-0.56, respectively. All the printing models were printed using a slice thickness of 100 μm. The exposure time was 10 s for each layer. A tumor printed by different

materials was assembled on the cavity by self-welding. Hollow hepatic veins and portal veins were filled by blue paste and green paste, respectively.

## Characterization

Mechanical tests were conducted at a strain rate of 100%/min, using universal tensile testing machine (SUNS). For curing kinetics, the double bond conversion was measured by real-time FT-IR spectrometer (Thermo Fisher Scientific, Nicolet, 5700). Hydrogen bond was characterized with temperature-variable FTIR spectrometer (Thermo Fisher Scientific, Nicolet, iS50). The spectra were recorded every 20 °C with 32 scans in absorption mode at resolution of 2 cm⁻¹. The self-healing process of a scratch was recorded by optical microscope (Zeiss, ZCEC-150348F). The rheological properties were characterized by HAAKE RS6000. Iso-strain stress relaxation and shape recovery were conducted by a dynamic mechanical analyzer (TA Q800).

## Reporting summary

Further information on research design is available in the Nature Portfolio Reporting Summary linked to this article.

## Data availability

The authors declare that the data supporting the findings of this study are available within the paper and its Supplementary Information files. Additional data are available from the corresponding author upon request. Source data are provided as a Source Data file. Source data are provided with this paper.

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

## Acknowledgements

T.X. thanks the following programs for the financial support: National Key R&D Program of China (No. 2022YFB3805701). Q.Z. thanks the fol-lowing programs for the financial support: National Natural Science Foundation of China (U20A6001 and 52273112), National Key R&D Pro-gram of China (2022YFA1103500).

## Author contributions

J.W., Y.L., and F.H. designed and conducted the experiments. J.W. and X.C. wrote the paper. Q.Z., T.X., and Y.Z. supervised the project. All authors contributed to the discussion.

## Competing interests

The authors declare no competing interests.

## Additional information

**Peer review information** *Nature Communications* thanks Huang Wei, Georgios Tsoulfas and the other, anonymous, reviewer(s) for their con-tribution to the peer review of this work. A peer review file is available.

