## [Peer Review File · Nature Communications]

3D printing of self-healing personalized liver models for surgical training and preoperative planningREVIEWER COMMENTS

Reviewer #1 (Remarks to the Author):

1. The authors report the self-healing elastomers via stereolithographic 3D printing from copolymerization of 4-acryloylmorpholine (ACMO) and methyl poly (ethylene glycol) acrylate (mPEGA). The elastomers exhibit tunable modulus and efficient self-healing capability. It is shown that the printed liver models based on the elastomers have liver-like modulus and can facilitate the achievement of tumor-free surgical margin and the protection of important anatomical structures via a trial-and-error method. This work is interesting and the result has promising application.
2. The innovation of this work is not so strong, as this kind of self-healing elastomers based on hydrogen bond interactions is widely reported.
3. The authors emphasize only the importance of liver-like modulus for liver models, except modulus, what other properties of the elastomers should be taken into account when used for liver modulus?
4. Methyl poly (ethylene glycol) acrylate (mPEGA) should be poly (ethylene glycol) mono-methacrylate (PEGMMA). The chemical structure of PEGMMA shown in Figure 1 (a) is wrong (methyl group is missing).

Reviewer #2 (Remarks to the Author):

This is an interesting paper evaluating the use of 3D printed liver models for surgical training and preoperative planning. The authors present their 3D printing technique using material that allows the liver to be self-healing, and thus able to be used for multiple training instances.

Could the authors please respond to the following questions:

- 1) To what degree of detail in terms of arterial, portal and biliary flow is the 3D printed liver accurate in the inside? That is, how detailed is the printed liver with this material in terms of the relation of the target lesion to the surrounding vascular and biliary structures?
- 2) The authors report the two cases here using the 3D printed liver. Do the authors have measurable/objective data in terms of how has the use of the 3D printed liver made the hepatic resection safer or easier or more efficient or easier to teach?
- 3) Could the authors discuss in more detail the limitations of the paper?

Reviewer #3 (Remarks to the Author):

Summary

=====

The present work deals with 3 d printed, self-healing elastomers, and shows a possible application case for preoperative planning.

This is a very interesting and promising work. It focuses more or less on three topics – material processing/printing, self-healing, and application on preoperative models. The processing/printing and self welding are recently presented in a similar form by the authors of this paper. The application for surgical training models on a human liver is original.

The basic question arises to whom this work is addressed (Chemical or biomedical engineers, medical experts), a clear focus and objective are missing.

The parts are not strongly connected and written differently. The novel part is the usage of the model for surgical training. Thus the review focuses mainly on that. If also the other parts contain novelty, it should be clearly stated how this work differs from previous studies.

Important information about the surgical model topic is missing, and more details would be helpful (see comments). Overall, this is a relevant work, but it is suggested to rethink the objectives as well as the focus and revising the work.

Detailed Comments

Models for surgical training: The authors focus here on 2 important features that such models should have – stiffness and self-healing. However, there are a number of other important aspects that have not been addressed here:

1)Stiffness: This aspect is much more complex than presented here. Liver is a non-linear viscous material that cannot be simply described with one tensile test and one elastic E-modulus (how was this measured and evaluated?). There are several way to describe such a material, but also viscous parameters are important that have not been mentioned. Is the new material rather visco-elastic-plastic or is it hyperelastic? Especially for the haptic behavior not only the initial compliance but also the stiffening behavior during “pressing” and the viscous behavior in terms of hysteresis is important for the “feeling”. These things

have been studied recently extensively, specifically on the liver, but not considered or mentioned here. It is recommended to do a comprehensive literature study in this direction. A cycling experiment should be done to show that this material is similar to liver and outperforms currently used materials like soft 3D printing material (e.g. from Stratasys) or silicones for such surgical models.

2) Pre-operative planning: Important requirements for such models are a simple/fast manufacturability, similar mechanical behavior as the biological tissue, but also the possibility of manipulation. In addition to cutting with a scalpel (presented here but not well described), electro-cutting, and suturing (pulling out) are also very important features. The question is how the new material performs in the latter two as well as how realistic is the cutting with a scalpel - is there an feedback from surgeon?

3) In this work, a model which is only 50% of the original size was used and there is no information about the real costs of a 3D print although the authors claim to produce a "similar to reality" and cost-effective model. A new feature of the material which is very interesting is the repetitive cutting option. However, in some cases this could be a disadvantage e.g. when slipping of surfaces is important like in joints.

4) In many cases of surgical models, 3D Printing is not the first method of choice, casting is much faster to fabricate big parts. 3D printing makes sense for tiny, hollow, micro-structured parts or vessels, etc. this aspect should be also discussed when talking about surgical training models.

5) How good is the material really compared to other materials? i.e. Hydrogel vs. Silicon vs. soft 3D printed material vs. real liver: It would be also interesting to see a comparison of typical artificial materials as well as a comparison to real liver tissue wrt to the above-mentioned points. Especially how about the handling properties (sticking to cloughs) and drying out of hydrogels.

6) Toxicity, environmental compatibility, and recycling: Could you give any statements on that?

L188: It is not clear how exactly the model was created. It seems individual parts are printed, and the model is assembled and filled with liquid. If the focus of a revised work is on surgical models, more details should be given in this part. One aspect not mentioned in

this context – how about self-healing of separately printed parts? There was work published which shows a different behavior if cut surfaces or other outer-surfaces are glued together.

I guess there are also limitations which should be described.

Reviewer #1 (Remarks to the Author):

The authors report the self-healing elastomers via stereolithographic 3D printing from copolymerization of 4-acryloylmorpholine (ACMO) and methyl poly (ethylene glycol) acrylate (mPEGA). The elastomers exhibit tunable modulus and efficient self-healing capability. It is shown that the printed liver models based on the elastomers have liver-like modulus and can facilitate the achievement of tumor-free surgical margin and the protection of important anatomical structures via a trial-and-error method. This work is interesting and the result has promising application.

Answer: We thank the reviewer for the positive comments.

Q1. The innovation of this work is not so strong, as this kind of self-healing elastomers based on hydrogen bond interactions is widely reported.

Answer: We thank the reviewer for the comment. Just as the reviewer said, self-healing elastomers based on hydrogen bond interactions have been widely reported. However, there is barely real practical application of self-healing materials reported. Also, most of the previous work focuses only on the mechanism and performance of the self-healing capacity. Our work aims at developing self-healing and re-cuttable liver models for surgical training and preoperative planning. So, several specialized criteria should be met. First, the elastomer should be self-healed efficiently at room temperature within several minutes. Second, considering the practicality, it is better to use commercialized raw materials in order to keep a low cost, complex synthesis process is not preferred here. Third, the precursor of the elastomer should have a low viscosity and a high reactivity to guarantee an efficient 3D printing. To the best of our knowledge, no previously reported self-healing elastomer can satisfy all these demands at the same time. In this work, two commercial monomers are used to get a liquid resin with high reactivity and ultra-low viscosity in the absence of any chemical crosslinkers. Indeed, such a DLP 3D printed thermoplastic self-healing elastomer has not been reported yet. Hydrogen bond interaction and the linear chain topology together contribute to the efficient self-healing at room temperature. We find that introduction of a small amount of chemical crosslinker or other monomers which can not form hydrogen bond with ACMO will both significantly decline the self-healing performance (Fig. 3c).

Q2. The authors emphasize only the importance of liver-like modulus for liver models, except modulus, what other properties of the elastomers should be taken into account when used for liver modulus?

Answer: We agree with the reviewer's comment. Indeed, liver is a non-linear viscous material that cannot be simply described with tensile test and elastic E-modulus. Viscoelastic properties of the self-healing elastomer have been systematically

investigated and added in the revised manuscript (Fig. 2e-h). Hysteretic curves, G'/G'' , stress relaxation, and shape recovery have been studied through cyclic tensile test, shear rheology test, and dynamic mechanical analysis. On the whole, all the results show that the self-healing elastomer is not a hyper-elastic material. Nonnegligible energy loss derived from the intrinsic viscosity of the material can be observed when it is deformed. This is coincident with the biomechanical characteristics of liver [1-3]. The viscoelasticity of the elastomers can be tuned through the adjustment of the composition to simulate a human liver as much as possible (Fig. 2i).

Q3. Methyl poly (ethylene glycol) acrylate (mPEGA) should be poly (ethylene glycol) mono-methacrylate (PEGMMA). The chemical structure of PEGMMA shown in Figure 1 (a) is wrong (methyl group is missing).

Answer: The chemical structure is right. The name of the reactant has been corrected into methoxy poly (ethylene glycol) acrylate (mPEGA).

Reviewer #2 (Remarks to the Author):

This is an interesting paper evaluating the use of 3D printed liver models for surgical training and preoperative planning. The authors present their 3D printing technique using material that allows the liver to be self-healing, and thus able to use for multiple training instances. Could the authors please respond to the following questions:

Answer: We thank the reviewer for the positive comments.

Q.1 To what degree of detail in terms of arterial, portal and biliary flow is the 3D printed liver accurate in the inside? That is, how detailed is the printed liver with this material in terms of the relation of the target lesion to the surrounding vascular and biliary structures?

Answer: We thank the reviewer for the comment. A top-down DLP process was chosen since it provides better printing stability (e. g. a good self-support for soft materials) in comparison to the bottom-up process, especially for large-sized objects. As for the printing resolution, pillars with diameters of 500 μm can be readily printed, which is a quite high printing resolution considering the softness of the ACEG-x elastomer and the large printable size of our 3D printer (380*240*100 mm). So, from the perspective of printing accuracy, theoretically, it is possible for us to print the majority of tissues in the liver, such as the abdominal cavity, the retrohepatic inferior vena cava, and the gall bladder. At the current stage, we focused on the hepatic veins and portal veins as they receive most attention in liver surgery. The main veins with their first and second lever branches were hollowed and filled (Fig. 4a). Thinner blood vessels and biliary tract were omitted.

Q.2 The authors report the two cases here using the 3D printed liver. Do the authors have measurable/objective data in terms of how has the use of the 3D printed liver made the hepatic resection safer or easier or more efficient or easier to teach?

Answer: We thank the reviewer for the comment. To explore the feasibility of 3D printed liver models for preoperative planning, a phase 1, preliminary interventional study is registered on ClinicalTrials.gov (ID: NCT06006338). In this study, a total of 5 models were employed by experienced surgeons for preoperative planning purposes. The highly personalized models can be printed soon after preoperative enhanced CT examination and perspective planning can be implemented on the model before the authentic surgery. According to surgeons' feedback, the addition of a physical model facilitated a clearer and more vivid comprehension of the personalized features of a tumor. More importantly, the try-and-error method allowed them to determine the optimal surgical plane. In the majority of cases, surgeries were successfully carried out without interruption, adhering to the pre-planned surgical approach. There was no need to pause the surgery, review CT/MRI data, or conduct intraoperative ultrasound, as had been required in previous operations. In a word, preoperative planning on models makes the hepatic resection safer by maximizing the degree of certainty and minimizing the possibility of emergency situations, which is hard to achieved by previously reported 3D printed models and other 3D visualization methods. However, it should note that this study did not include a control group because the clinical outcomes are difficult to measure quantitatively. Prospective randomized controlled trails will be addressed in future research endeavors.

Q.3 Could the authors discuss in more detail the limitations of the paper?

Answer: In this work, we try to propose a 3D printed precise liver model which exhibit a repetitive cutting capability to be applied in surgical training and personalized preoperative planning. To achieve this goal, we develop a low-cost, highly reactive, self-healing elastomer with tailorable mechanical properties. Preliminary trials in clinical practice demonstrate that this self-healing model allows surgeons to determine optimal operation traces via repeating surgical simulations on the printed models. However, just as proposed by the reviewers, the previous version of the manuscript does have limitations in the aspects of material, process, and application.

1. We emphasized only the importance of modulus for liver models, and failed to notice that the liver is a non-linear viscous material. We have systematically investigated the viscoelasticity of the material and added the results in the revised manuscript. The results show that the mechanical properties of the self-healing elastomer can be easily tailored to get close to the real biomechanical characteristics of the liver.

2. Limited by the available printing size of our 3D printer, we can only print a half-size liver model. We have already started to customize a larger DLP 3D printer, but it will take some time.

3. Considering the difficulty of manufacturing in the present experimental conditions, the model is simplified, and only includes liver, tumor, hepatic vein, and portal vein. If

other structures (such as the abdominal cavity, the retrohepatic inferior vena cava, and the gall bladder) can be produced and assembled together with the printed liver model, this technology will be more in line with real-life surgical setting and have broader clinical prospects. With the improvement in printing accuracy of 3D printers and the optimization of our materials, I believe that this problem can be resolved in the near future.

Reviewer #3 (Remarks to the Author):

Summary

=====

The present work deals with 3d printed, self-healing elastomers, and shows a possible application case for preoperative planning.

This is a very interesting and promising work. It focuses more or less on three topics – material processing/printing, self-healing, and application on preoperative models. The processing/printing and self-welding are recently presented in a similar form by the authors of this paper. The application for surgical training models on a human liver is original.

The basic question arises to whom this work is addressed (Chemical or biomedical engineers, medical experts), a clear focus and objective are missing.

The parts are not strongly connected and written differently. The novel part is the usage of the model for surgical training. Thus, the review focuses mainly on that. If also the other parts contain novelty, it should be clearly stated how this work differs from previous studies.

Important information about the surgical model topic is missing, and more details would be helpful (see comments). Overall, this is a relevant work, but it is suggested to rethink the objectives as well as the focus and revising the work.

Detailed Comments

Models for surgical training: The authors focus here on 2 important features that such models should have – stiffness and self-healing. However, there are a number of other important aspects that have not been addressed here:

Q1. Stiffness: This aspect is much more complex than presented here. Liver is a non-linear viscous material that cannot be simply described with one tensile test and one elastic E-modulus (how was this measured and evaluated?). There are several ways to describe such a material, but also viscous parameters are important that have not been mentioned. Is the new material rather visco-elastic-plastic or is it hyperelastic? Especially for the haptic behavior not only the initial compliance but also the stiffening behavior during “pressing” and the viscous behavior in terms of hysteresis is important for the “feeling”. These things have been studied recently extensively, specifically on

the liver, but not considered or mentioned here. I recommend to do a comprehensive literature study in this direction. A cycling experiment should be done to show that this material is similar to liver and outperforms currently used materials like soft 3D printing material (e.g. from Stratasys) or silicones for such surgical models.

Answer: We agree with the reviewer's comment. We admit that it is not enough to emphasize only the importance of liver-like modulus for liver models. Other properties of the elastomers should be considered when used for liver modulus. According to the reviewer's suggestion, viscoelastic properties of the self-healing elastomer have been systematically investigated and added in the revised manuscript (Fig. 2e-h). Hysteretic curves, G'/G'' , stress relaxation, and shape recovery have been studied through cyclic tensile test, shear rheology test, and dynamic mechanical analysis. On the whole, all the results show that the self-healing elastomer is not a hyper-elastic material. Nonnegligible energy loss derived from the intrinsic viscosity of the material can be observed when it is deformed. This is coincident with the biomechanical characteristics of liver^[1-3]. The viscoelasticity of the elastomers can be tuned through the adjustment of the composition to simulate a human liver as much as possible (Fig. 2i).

Q2. Pre-operative planning: Important requirements for such models are a simple/fast manufacturability, similar mechanical behavior as the biological tissue, but also the possibility of manipulation. In addition to cutting with a scalpel (presented here but not well described), electro-cutting, and suturing (pulling out) are also very important features. The question is how the new material performs in the latter two as well as how realistic is the cutting with a scalpel - is there a feedback from surgeon?

Answer: We thank the reviewer for the comment. The ultrasound scalpel is one of the most commonly used cutting device, and is routinely used to resect a tumor from the liver by the surgeons of our team in their center. The active blade of an ultrasonic scalpel can make ultra-high frequency oscillations and produce frictional heat, so that liver parenchyma is divided and the container vessels are sealed. The material used in this study can be easily cut by an ultrasound scalpel, and we have compared the changes of related parameters when the material was cut using an ordinary scalpel and an ultrasound scalpel (Fig. 4b). It turned out that changes were similar in both conditions. Besides, according to the feedback of surgeons in our team, it brought similar feels in terms of cutting resistance or heat changes when they cut the liver parenchyma of a patient and the printed model with an ultrasound scalpel. They also experienced analogous feeling when making suturing movements in the liver parenchyma of a patient and the printed model.

Q3. In this work, a model which is only 50% of the original size was used and there is no information about the real costs of a 3D print although the authors claim to produce a "similar to reality" and cost-effective model. A new feature of the material which is very interesting is the repetitive cutting option. However, in some case this could be a disadvantage e.g. when slipping of surfaces is important like in joints.

Answer: We thank the reviewer for the comment. It is a great pity that we can only print a half-size liver model at this stage. This is limited by the available printing size of our 3D printer (380*240*100 mm). We have already started to customize a larger DLP 3D printer, but it will take some time.

The total cost of a 3D liver model includes the cost of material, 3D reconstruction and modeling, and 3D printing. Benefiting from the absence of any complex synthesis process, the cost of the resin can be as low as 10 dollar per kilogram. DLP 3D printing is a relatively low-cost printing technique comparing with other techniques such as Polyjet. Considering of the depreciation of the printer, the printing cost can be controlled to 2 dollars. The 3D reconstruction will cost 600 dollars per part. However, the 3D reconstruction is routinely performed for the necessity of surgery planning in the hepatobiliary center of surgeons in our team. Namely, the 3D construction is not purposely performed for our study. So, the total cost of a personalized liver model will be less than 20 dollars, which we believe is acceptable.

In this work, we focus mainly on the self-healing properties of the material in order to realize repetitive cutting for surgical training and personalized preoperative planning. We should admit that the materials reported in this work can't be used to simulate all the tissues or organs in regard of the different biomechanical characteristics. Specialized materials should be developed for different purposes.

Q4. In many cases of surgical models, 3D Printing is not the first method of choice, casting is much faster to fabricate big parts. 3D printing makes sense for tiny, hollow, micro-structured parts or vessels, etc. this aspect should be also discussed when talking about surgical training models.

Answer: We thank the reviewer for the comment. Just as the reviewer said, 3D printing is more suitable to fabricate complex parts with internal cavities or lattice structures. In fact, sacrificial template or mode are always 3D printed and then casted into personalized surgical models. Directly 3D printing surgical models is a more efficient way, however, greater demands are being placed on the materials performance. Here, we first 3D print a liver model with internal hollow vessels, and then acrylic paints with different colors were injected to distinguish between different vessels (hepatic vessel and portal vessel) and the liver tissue. It will be a really difficult to fabricate such a multi-material model using the casting strategy.

Q5. How good is the material really compared to other materials? i.e. Hydrogel vs. Silicon vs. soft 3D printed material vs. real liver: It would be also interesting to see a comparison of typical artificial materials as well as a comparison to real liver tissue wrt to the above-mentioned points. Especially how about the handling properties (sticking to cloughs) and drying out of hydrogels.

Answer: We thank the reviewer for the comment. Indeed, silicon is a good choice for the preparation of liver model. However, 3D printing of a high-precision model with

DLP technique is difficult and expensive due to the lack of commercially-available photo-reactive raw materials. Comparing to the self-healing ACEG elastomers we proposed, hydrogels have a much worse mechanical robustness and stability. The mechanical properties of other available commercial soft 3D printing materials are too limited to simulate the liver. In addition, there is still great challenge to introduce the efficient room temperature self-healing capability to the 3D printable silicon, hydrogels, and other commercial soft materials in a low-cost way. As a result, we believe that our approach using self-healing ACEG elastomers shows great promise in improving the current state of personalized liver models for surgical training.

Q6. Toxicity, environmental compatibility, and recycling: Could you give any statements on that?

Answer: We thank the reviewer for the constructive comment. Majorities of the existing materials for DLP or SLA printing are crosslinked thermosets, which have already been an environmental issue due to its difficulty in recycling. Absence of chemical crosslinkers within the self-healing elastomer means that the abandoned surgical models can be easily recycled like other thermoplastic products.

In regard to the toxicity, both of the two raw materials, AMCO and mPEGA, exhibit low odor and low skin irritation.

Q7. L188: It is not clear how exactly the model was created. It seems individual parts are printed, and the model is assembled and filled with liquid. If the focus of a revised work is on surgical models, more details should be given in this part. One aspect not mentioned in this context – how about self-healing of separately printed parts? There was work published which shows a different behavior if cut surfaces or other outer-surfaces are glued together.

Answer: We thank the reviewer for the comment, and we are sorry that the fabrication process was not presented clearly. A more detailed process is now added in the manuscript (Fig. 4a). The tumor (ACEG-0.56, stiffer) and the liver with hollow vessels (ACEG-0.50, softer) are individually printed and assembled benefiting from the self-healing capability. Then, acrylic paints with different colors were injected to distinguish between different vessels (hepatic vessel and portal vessel) and the liver tissue.

Self-healing (self-welding) of separately printed parts have been evaluated and demonstrated in Fig. 3e. The self-healing performance is comparable to that of newly-cut surfaces (Fig. 3a, b). We attribute the good self-healing to the absence of chemical crosslinking. The polymer chain segments are more prone to migration, diffusion, and entangling, which is favorable for the crack healing and self-welding.

Reference:

[1] Mattei, G., Ahluwalia, A., Sample, testing and analysis variables affecting liver

mechanical properties: A review. *Acta Biomater.* **45**, 60-71 (2016).

[2] Estermann, S., Förster-Streffleur, S., Hirtler, L., Streicher, J., Pahr, D., Reisinger, A., Comparison of Thiel preserved, fresh human, and animal liver tissue in terms of mechanical properties. *Ann. Anat.* **236**, 151717 (2021).

[3] Marchesseau, S., Heimann, T., Chatelin, S., Willinger, R., Delingette, H., Fast porous visco-hyperelastic soft tissue model for surgery simulation: Application to liver surgery. *Prog. Biophys. Mol. Bio.* **103**, 185-196 (2010).

REVIEWERS' COMMENTS

Reviewer #1 (Remarks to the Author):

The authors have replied my questions very carefully and revised the manuscript correspondingly. They explained the innovation points of this work in detail and provided the investigation on the viscoelastic properties of the self-healing elastomer in the revised manuscript. I have no further question.

Reviewer #2 (Remarks to the Author):

I would like to thank the authors for their responses and the changes made accordingly

Reviewer #3 (Remarks to the Author):

All in all, the authors have significantly revised and improved the manuscript. They added new experimental data, improved pictures, and responded to my comments.

Still, the objectives in the abstract could be slightly improve by inserting into Line 29 a sentence like (or similar): ... for demonstration purposes, ... liver models were printed ...

Otherwise, I agree with the present content of the manuscript.